# LargeST: A Benchmark Dataset for Large-Scale Traffic Forecasting

**Xu Liu[1], Yutong Xia[1], Yuxuan Liang[2]\*, Junfeng Hu[1], Yiwei Wang[1], Lei Bai[3],**
**Chao Huang[4], Zhenguang Liu[5]\*, Bryan Hooi[1], Roger Zimmermann[1]**
[1]National University of Singapore, [2]Hong Kong University of Science and Technology (Guangzhou),
[3]Shanghai AI Laboratory, [4]University of Hong Kong, [5]Zhejiang University
{liuxu, junfengh, y-wang, bhooi, rogerz}@comp.nus.edu.sg, {yutong.x, yuxliang}@outlook.com,
{baisanshi, chaohuang75, liuzhenguang2008}@gmail.com

## Abstract

Road traffic forecasting plays a critical role in smart city initiatives and has experienced significant advancements thanks to the power of deep learning in capturing non-linear patterns of traffic data. However, the promising results achieved on current public datasets may not be applicable to practical scenarios due to limitations within these datasets. First, the limited sizes of them may not reflect the real-world scale of traffic networks. Second, the temporal coverage of these datasets is typically short, posing hurdles in studying long-term patterns and acquiring sufficient samples for training deep models. Third, these datasets often lack adequate metadata for sensors, which compromises the reliability and interpretability of the data. To mitigate these limitations, we introduce the LargeST benchmark dataset. It encompasses a total number of 8,600 sensors in California with a 5-year time coverage and includes comprehensive metadata. Using LargeST, we perform in-depth data analysis to extract data insights, benchmark well-known baselines in terms of their performance and efficiency, and identify challenges as well as opportunities for future research. We release the datasets and baseline implementations at: `https://github.com/liuxu77/LargeST`.

## 1 Introduction

Road traffic forecasting plays a pivotal role in enhancing urban planning, traffic management, and public safety, making it one of the most critical components of Intelligent Transportation Systems [35]. In recent years, the burgeoning technique of machine learning, particularly deep learning, has exhibited remarkable advancements in this task [26]. Among the advances, spatio-temporal graph neural networks have demonstrated great promise and become the widely embraced tool for accurate traffic predictions [34, 19, 32, 1, 4]. Specifically, they utilize graph neural networks to capture spatial correlations among sensors and employ sequential models to model temporal dependencies.

What steps can we take to further advance research in traffic forecasting? Historically, high-quality and large-scale benchmark datasets have proven their value in driving research frontiers, as exemplified by the transformative impact of ImageNet [6] in computer vision, GLUE [31] in natural language processing, and OGB [13] in the general graph. In traffic forecasting research, however, we argue that commonly-used datasets present critical issues that may pose obstacles to future progress.

---

\*Y. Liang and Z. Liu are the corresponding authors of this paper.

37th Conference on Neural Information Processing Systems (NeurIPS 2023) Track on Datasets and Benchmarks.

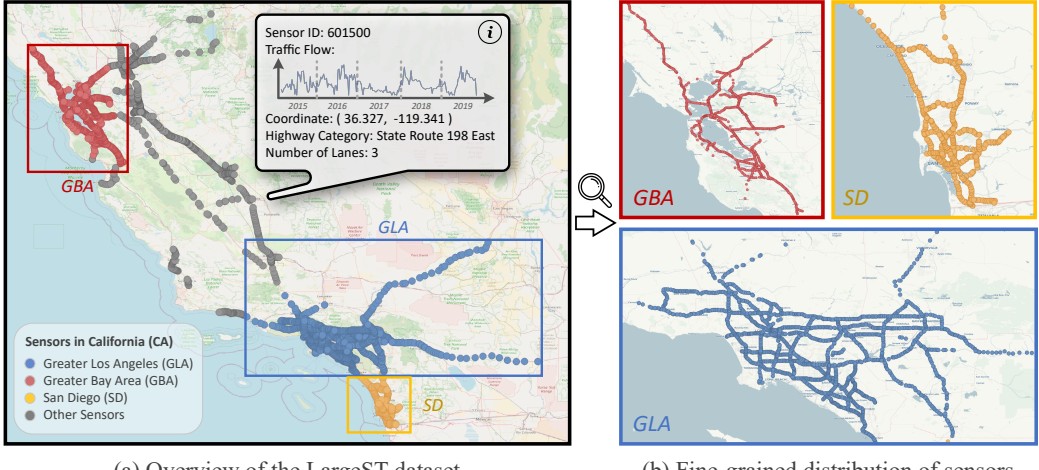

(a) Overview of the LargeST dataset        (b) Fine-grained distribution of sensors

Figure 1: An illustration of the LargeST benchmark dataset.

**Issues of Existing Datasets**. Most of the widely-used traffic datasets are limited in scale compared to real-world traffic networks. For example, widely-utilized benchmarks such as PeMS03, 04, 07, and 08 [30], comprise merely hundreds of nodes and edges (see Table 1 for details). However, the real-world traffic networks usually have much larger scales to be analyzed. For instance, California, US, alone possesses nearly 20,000 operational sensors. As traffic forecasting models are extensively developed on these small datasets, the majority of them turn out to be not scalable to larger sensor networks (see Section 5). Moreover, existing traffic datasets suffer from a dearth of temporal coverage, often spanning less than 6 months of data. This restricted duration hinders the study of long-term seasonal patterns and limits the number of training instances that can be used for deep model training. Another critical limitation of commonly-used datasets lies in the insufficient metadata available for individual nodes. Considering that traffic sensors represent specific points with tangible significance, the incorporation of node metadata in datasets is crucial. This inclusion serves to enhance the overall reliability of datasets and enables better interpretability of model predictions.

**Contributions**. In this work, we propose LargeST as a new benchmark dataset (see Figure 1), with the goal of facilitating the development of accurate and efficient methods in the context of large-scale traffic forecasting. The distinguishing characteristic of LargeST lies not only in its extensive graph size, encompassing a total of 8,600 sensors in California, but also in its substantial temporal coverage and rich node information – each sensor contains 5 years of data and comprehensive metadata. In addition to dataset construction, we conduct comprehensive data analysis, implement a suite of well-known traffic forecasting baselines, and perform extensive benchmark experiments. According to the empirical results, we summarize and highlight research challenges and future opportunities. LargeST is an open-source project hosted on GitHub. We will keep monitoring new research advancements in the field and summarize them in the repository.

## 2 Preliminaries

This section commences by introducing the traffic forecasting task, followed by a brief review of existing efforts in this area.

### 2.1 Problem Statement

The objective of traffic forecasting is to predict target attributes (e.g. traffic flow) in future steps based on historical observations over a directed sensor graph. To define the graph topology, the common practice [34, 19, 32] build the adjacency matrix $A$ using a thresholded Gaussian kernel [29], where $A_{ij} = \exp\left(-\frac{d_{ij}^2}{\sigma^2}\right)$ if $\exp\left(-\frac{d_{ij}^2}{\sigma^2}\right) \geqslant r$ else $A_{ij} = 0$. Here, $d_{ij}$ denotes the road network distance between sensors $i$ and $j$, $\sigma$ is the standard deviation of all distances, and $r$ is the threshold.

## 2.2 Deep Learning-based Traffic Forecasting

In recent years, deep neural networks have emerged as the preferred method for traffic forecasting [19, 2, 22, 23, 7], thanks to their advanced learning capacity. They generally combine graph neural networks (GNNs) with either Recurrent Neural Networks (RNNs) or Temporal Convolutional Networks (TCNs) to capture intricate spatial and temporal dependencies in traffic data. For example, a pioneering work DCRNN [19] introduced a novel diffusion convolution that works alongside GRU. Other notable works such as ST-MetaNet [27], AGCRN [1], and DGCRN [18] have also employed RNNs and their variants. To improve training speed and leverage parallel computation, a plethora of approaches such as STGCN [34], GWN [32], and DMSTGCN [11] have replaced RNNs with dilated causal convolution for temporal patterns modeling. Moreover, attention mechanisms have been utilized in works like GeoMAN [20] and ASTGCN [10] to model spatial and temporal correlations. Recent developments in the field highlight two noteworthy trends. The first involves coupling GNNs with neural ordinary differential equations to produce continuous layers for modeling long-range spatio-temporal dependencies, e.g., STGODE [8] and STGNCDE [4]. The second trend involves building adjacency matrices at different time steps to capture the dynamic correlations among nodes, e.g., DGCRN [18], DSTAGNN [17], and D$^2$STGNN [28]. These approaches have exhibited promising performance on existing datasets, which, however, have limited size in terms of the number of nodes, edges, and time frames (see Table 1). This raises valid concerns about their scalability when applied to large-scale traffic forecasting datasets, which strive to encompass more realistic real-world scenarios.

## 3 Limitations of Existing Traffic Datasets

Table 1 presents a comparison between the proposed LargeST and other popular traffic datasets. We next detail the improvements of LargeST over others from three aspects.

**Larger Graph Size**. LargeST stands out from previous traffic datasets in terms of graph size. For instance, CA includes $8.4\times - 50.6\times$ more nodes and $13.9\times - 729.6\times$ more edges than its predecessors. Regarding the graph structure, we observe that the sensor graphs released by [30] are extremely sparse, with an average degree of only around 1. This high degree of sparsity significantly limits the effectiveness of graph-based models in capturing the spatial correlations among nodes. Although the situation in the other two data sources is better, they are still not comparable to CA. In short, the increased graph size of LargeST provides a more realistic depiction of the scale of road networks, and offers an excellent testing ground for evaluating the scalability of both current and future traffic forecasting models.

**Higher Temporal Coverage**. The temporal coverage of existing traffic datasets is often limited, typically spanning no more than 6 months. In contrast, LargeST covers an unprecedented 5 years of traffic data with the same sampling rate as previous datasets, offering several potential benefits. First, it allows for the study of long-term patterns, such as seasonal trends on a weekly or monthly basis. Second, compared to some datasets that date back to 2012, LargeST is more up-to-date and thus better reflects recent traffic conditions. Third, the inclusion of a longer time period provides a larger sample size for model training, which is especially advantageous for deep learning models and becomes even more crucial in the current data-centric AI landscape.

**Richer Node Metadata**. Existing datasets often lack sufficient metadata for nodes, with some even omitting this important information entirely. This limitation greatly cripples the reliability of the data, since users are left unaware of the precise locations of sensors, impeding their ability to effectively analyze and interpret data and model predictions. Moreover, we argue that traffic forecasting benefits from the provided node features in designing models. For example, node-level information such as the county or highway of a sensor can aid in node clustering [27], which could be helpful in the context of large-scale datasets. In addition, the number of lanes and variability of lane numbers along highways can significantly impact the frequency of accidents [15], which can have a direct effect on traffic flow readings (see Section 4.2.3 for more details).

Table 1: Comparisons between LargeST and other popular datasets. Degree: the average degree of each node. Meta: the number of metadata associated with each node. Data Points: multiplication of nodes and frames. M: million ($10^6$). B: billion ($10^9$). The sampling rate for all datasets is at 5 minutes level.

| Released Source | Dataset | Nodes | Edges | Degree | Meta | Time Range | Frames | Data Points |
|---|---|---|---|---|---|---|---|---|
| Yu et al. [34] | PeMSD7(M) | 228 | 1,664 | 7.3 | 6 | 05/01/2012 – 06/30/2012 | 12,672 | 2.89M |
| | PeMSD7(L) | 1,026 | 14,534 | 14.2 | 0 | 05/01/2012 – 06/30/2012 | 12,672 | 13.00M |
| Li et al. [19] | METR-LA | 207 | 1,515 | 7.3 | 3 | 03/01/2012 – 06/27/2012 | 34,272 | 7.09M |
| | PEMS-BAY | 325 | 2,369 | 7.3 | 3 | 01/01/2017 – 06/30/2017 | 52,116 | 16.94M |
| Song et al. [30] | PEMS03 | 358 | 546 | 1.5 | 1 | 09/01/2018 – 11/30/2018 | 26,208 | 9.38M |
| | PEMS04 | 307 | 338 | 1.1 | 0 | 01/01/2018 – 02/28/2018 | 16,992 | 5.22M |
| | PEMS07 | 883 | 865 | 1.0 | 0 | 05/01/2017 – 08/06/2017 | 28,224 | 24.92M |
| | PEMS08 | 170 | 276 | 1.6 | 0 | 07/01/2016 – 08/31/2016 | 17,856 | 3.04M |
| LargeST (ours) | CA | 8,600 | 201,363 | 23.4 | 9 | 01/01/2017 – 12/31/2021 | 525,888 | 4.52B |
| | GLA | 3,834 | 98,703 | 25.7 | 9 | 01/01/2017 – 12/31/2021 | 525,888 | 2.02B |
| | GBA | 2,352 | 61,246 | 26.0 | 9 | 01/01/2017 – 12/31/2021 | 525,888 | 1.24B |
| | SD | 716 | 17,319 | 24.2 | 9 | 01/01/2017 – 12/31/2021 | 525,888 | 0.38B |

## 4 The LargeST Benchmark Dataset

This section formally introduces the proposed LargeST benchmark dataset, which aims to comprehensively evaluate the accuracy, efficiency, and scalability of current and future traffic forecasting models in large-scale scenarios. We start by providing details on how LargeST is collected and organized in Section 4.1. We then conduct a thorough data analysis to gain a deeper understanding of LargeST in Section 4.2, and specify the licenses of data and code in Section 4.3.

### 4.1 Data Collection and Organization

As shown in Table 1, LargeST comprises four sub-datasets, each characterized by a different number of nodes. These sub-datasets collectively form a hierarchical structure, enabling the evaluation of models at various scales of nodes. We detail the sub-datasets as follows.

We source LargeST from the California Department of Transportation (CalTrans) Performance Measurement System [2] (PeMS) [3]. PeMS is an online platform that offers real-time traffic data collected from 18,954 loop detectors (sensors) across the California state highway system. To ensure our dataset represents the overall traffic conditions throughout the entire system, we specifically select sensors labeled as "mainline". We also exclude sensors that lack coordinate information or are extremely far away from other sensors. As a result, we obtain a dataset comprising a total of 8,600 sensors, which we refer to as California (CA).

To undertake a more meticulous analysis of traffic patterns across diverse regions of California, we construct three subsets of CA by selecting three representative areas within CA. The first one is GLA, which contains 3,834 sensors installed in 5 counties of the Greater Los Angeles area: Los Angeles, Orange, Riverside, San Bernardino, and Ventura. The second sub-dataset GBA, includes 2,352 sensors in 11 counties situated in the Greater Bay Area: Alameda, Contra Costa, Marin, Napa, San Benito, San Francisco, San Mateo, Santa Clara, Santa Cruz, Solano, and Sonoma. The smallest sub-dataset SD, comprises 716 sensors only in San Diego county. In addition to the county information, we provide other meta knowledge for each node, including their coordinates, district in PeMS, the highway they are located on, directions of travel, and number of lanes.

Moreover, to build the adjacency matrix of the sensor graph, we follow a common approach used in the field [34, 19, 25], which involves using road network distances. Specifically, we leverage the Open Source Routing Machine [3] [24], a high-performance routing engine run on OpenStreetMap [4] data, to query the shortest driving distance between sensors based on their coordinates. However,

---

[2]https://pems.dot.ca.gov
[3]https://project-osrm.org
[4]https://www.openstreetmap.org/

computing pairwise road network distances can be prohibitively time-consuming when dealing with a large number of nodes. To overcome this, we first compute pairwise geodesic distances between sensors, which is significantly faster than calculating the shortest path between them. Then we limit each node to only query its road network distances to other nodes that are within a 4-kilometer radius. Lastly, we follow Li et al. [19] to normalize the adjacency matrix by setting a small threshold that eliminates weak node connections. In this study, we opt to use a threshold of 0.01 to prune the connections, while ensuring an adequate number of edges are retained. However, the selection of this threshold is adaptable, and users may employ alternative values that align with their specific needs.

Furthermore, LargeST contains five years of traffic flow data (from 2017 to 2021) with a 5-minute interval (same as PeMS), resulting in a total of 525,888 time frames. In this study, we opt not to remove nodes with a high rate of missing traffic flow values. Because we intend to maintain the data in its original state as much as possible, so that users have the discretion to decide whether to fill missing values or not. Moreover, there exist a number of tools/methods to fill the missing values in the time series data. For example, the *interpolate()* function in the Pandas library is a handy and simple tool to do this.

## 4.2 Data Analysis

In this part, we conduct a thorough data analysis to unravel the intricate connections between traffic flow and various factors, including space, time, and meta features. We take the traffic data from 2019 as an illustrative example, to reduce the impact of the COVID pandemic.

### 4.2.1 Regional Disparities

In Figure 2 (a) and (b), for each sub-dataset, we first average the traffic flow across the spatial dimension at each time step, and further partition the data according to weekdays and weekends. In particular, the curve of CA serves as a baseline for the traffic flow level in the state. It is evident that the other three sub-datasets generally demonstrate higher flow values compared to CA. This observation is reasonable since these three sub-datasets correspond to major regions within CA, and other regions in CA typically have lower traffic volume. Comparing the patterns among GLA, GBA, and SD, we observe that they exhibit differences in both flow values and the shape of the curves, particularly on weekdays. These regional disparities can be ascribed to various factors such as differences in economic development levels, area topography, and road planning strategies.

### 4.2.2 Temporal Dynamics

We continue analyzing Figure 2 (a) and (b) from a temporal standpoint. First, on weekdays, the traffic flow exhibits two prominent peaks, with a morning peak around 8 a.m. and an evening peak typically occurring around 5 p.m. Second, there are notable discrepancies in the daily flow patterns between weekdays and weekends, attributed to the factors of diverse commuting routines, work schedules, and recreational pursuits. These observations underscore the significance of incorporating temporal features like time of day and day of week when designing models.

We next examine potential monthly patterns by visualizing the traffic flow distribution during the morning peak (7 a.m. – 8 a.m.) and evening peak (5 p.m. – 6 p.m.) in CA across 12 months in a year, as depicted in the box plots in Figure 2 (c) and (d), respectively. The results reveal two

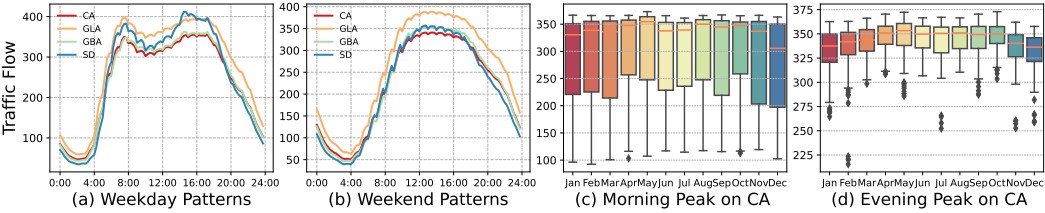

Figure 2: Relations between traffic flow and the factors in space and time.

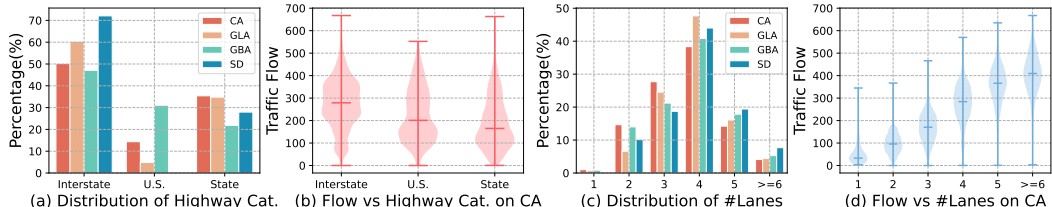

Figure 3: Relations between traffic flow and two crucial meta features: the highway categories and the number of lanes. The subplots (a) and (c) present the distribution of the features across four datasets. The subplots (b) and (d) show the violin plots of the average traffic flow of the CA dataset categorized by these two features.

noteworthy observations. (1) The morning peak displays a larger variance in traffic flow compared to the evening peak. This disparity could be attributed to factors such as flexible working hours in the morning or substantial traffic congestion in the evening. (2) Both peaks undergo a distribution shift throughout the months. This shift is characterized by a gradual increase from January to May, followed by fluctuations from June to September, and subsequently a decline from October to December. This observation aligns with typical human mobility patterns during summer and winter seasons, with increased outdoor activities in summer and reduced activities in winter, and suggests that incorporating the month in year feature could be advantageous for improving model predictions. This also underscores the importance of training and evaluating models using wider temporal ranges, to ensure that they can generalize to such seasonally varying patterns.

### 4.2.3 Metadata Characteristics

In this part, we focus on analyzing the relationships between traffic flow and two key features: the highway categories and the number of lanes. In Figure 3, we visualize the distribution of two features across four datasets, along with the average traffic flow of sensors in the CA dataset categorized based on the features.

**Highway Categories**. There are three main kinds of highways in the United States[5], namely interstate highways, U.S. highways, and state highways. Different types of highways have different road designs, speed limits, and access points, thus impacting traffic patterns. In Figure 3 (a), we can find that the interstate is the most common type while U.S. highway is generally the least common one across four datasets. This aligns with the fact that U.S. highway is an older system and has been gradually replaced by interstate and state routes.

According to Figure 3 (b), which illustrates the average traffic flow of the CA dataset, we observe that interstate highways exhibit the highest average volume, followed by U.S. highways and state highways. This trend aligns with the distinct characteristics of each highway system. Specifically, interstate highways serve as essential transportation arteries that connect major cities, facilitating high volumes of vehicles for daily commutes, long-distance travel, and freight transport. U.S. highways exhibit slightly lower average volumes compared to interstate highways, as they primarily cater to intrastate travel and regional connectivity. State Highways, on the other hand, have the lowest average traffic volume among the three types, due to their localized scope and limited coverage.

**Factor of Lanes**. Figure 3 (c) presents the distribution of the number of lanes across four datasets, revealing that four lanes are the most prevalent configuration. Moreover, in Figure 3 (d), we show that there exists a positive correlation between the number of lanes on roads and the corresponding traffic volumes. The underlying rationale is as follows. The number of lanes on a road can affect traffic flow by influencing its capacity. Roads with more lanes generally have a higher capacity to accommodate a larger volume of vehicles, allowing for smoother traffic flow and reducing congestion during peak hours [16]. On the other hand, it can also affect traffic flow by influencing drivers' lane-changing behavior. With more lanes available, drivers have more opportunities to switch lanes, which can lead

---

[5]https://en.wikipedia.org/wiki/Numbered_highways_in_the_United_States

to disruptions in traffic flow. Frequent lane changes can cause merging conflicts, abrupt deceleration or acceleration, and an increased likelihood of accidents [15].

In short, the metadata exhibits strong correlations with traffic flow values, suggesting that they can serve as valuable external knowledge for improving predictive performance.

### 4.3 LargeST License

The LargeST benchmark dataset is released under a CC BY-NC 4.0 International License: `https://creativecommons.org/licenses/by-nc/4.0`. Our code implementation is released under the MIT License: `https://opensource.org/licenses/MIT`. The license of any specific baseline methods used in our codebase should be verified on their official repositories.

## 5 Experiments

### 5.1 Experimental Setup

**Datasets**. We conduct experiments on all sub-datasets of LargeST with the setting of predicting the 12-step future based on the 12-step historical data [19, 32]. To facilitate the study of traffic forecasting in a more extended future time period, we aggregate the traffic readings from 5-minute intervals into 15-minute windows, resulting in 96 time steps per day. Moreover, we use one year of traffic data from 2019 in experiments to allow for several less efficient baselines to be compared. We chronologically split the data into train, validation, and test sets, with a ratio of 6:2:2 for all sub-datasets, resulting in sample sizes of 21010, 7003, and 7004, respectively.

**Baselines**. We adopt the following representative traffic forecasting baselines. Historical Last (HL) [21] is a naive method that simply uses the last observation as all the future predictions. LSTM [12] is a temporal-only deep model that does not consider the spatial correlations. Taking advantage of the advancements in GNNs [5, 14], sequential models have been integrated with GNNs to effectively model traffic data. During the period from 2018 to 2020, we specifically select RNN-based methods such as DCRNN [19] and AGCRN [1], TCN-based methods like STGCN [34] and GWNET [32], as well as attention-based methods ASTGCN [10] and STTN [33]. Moreover, we incorporate four representative methods from the years 2021 and 2022, which reflect the recent research directions in the field. STGODE [8] leverages neural ordinary differential equations to effectively model the continuous changes of traffic signals. DSTAGNN [17], DGCRN [18], and D$^2$STGNN [28] specifically consider the dynamic characteristics of correlations among sensors on traffic networks.

**Implementation Details**. We collect the code of baselines from their respective GitHub repositories, perform necessary cleaning, and integrate them into a single repository to improve the reproducibility and ease of comparison. For the model and training related configurations, we follow the recommended settings provided in their code. Experiments are repeated twice with different seeds on an Intel(R) Xeon(R) Gold 6140 CPU @ 2.30 GHz, 376 GB RAM computing server, equipped with an NVIDIA RTX A6000 GPU with 48 GB memory.

**Evaluation Metrics**. We conduct a comprehensive comparison of baselines from the following aspects: (1) *Performance*. We assess the performance of the models using three commonly adopted metrics in forecasting tasks: mean absolute error (MAE), root mean squared error (RMSE), and mean absolute percentage error (MAPE). (2) *Efficiency*. We consider the efficiency of the models by measuring both the training and inference wall-clock time. We also report the batch size used during training to reflect their ability to handle large-scale datasets. Note that we specify a maximum batch size of 64. If a model is unable to run with this setting, we progressively decrease the batch size until it fully occupies the memory on an A6000 GPU.

### 5.2 Performance Comparisons

Table 2 presents the test results of MAE, RMSE, and MAPE for specific horizons of 3, 6, and 12, as well as the average values across all predicted horizons. Simple methods such as HL and LSTM

Table 2: Performance comparisons. We bold the best-performing baseline result. The absence of baselines on the GLA and CA datasets indicates that the models incur out-of-memory issue even when we set batch size to 4 on an A6000 GPU with 48 GB memory. Param: the number of learnable parameters. K: kilo ($10^3$). M: million ($10^6$).

| Data | Method | Param | Horizon 3 | | | Horizon 6 | | | Horizon 12 | | | Average | | |
|---|---|---|---|---|---|---|---|---|---|---|---|---|---|---|
| | | | MAE | RMSE | MAPE | MAE | RMSE | MAPE | MAE | RMSE | MAPE | MAE | RMSE | MAPE |
| SD | HL | – | 33.61 | 50.97 | 20.77% | 57.80 | 84.92 | 37.73% | 101.74 | 140.14 | 76.84% | 60.79 | 87.40 | 41.88% |
| | LSTM | 98K | 19.03 | 30.53 | 11.81% | 25.84 | 40.87 | 16.44% | 37.63 | 59.07 | 25.45% | 26.44 | 41.73 | 17.20% |
| | DCRNN | 373K | 17.14 | 27.47 | 11.12% | 20.99 | 33.29 | 13.95% | 26.99 | 42.86 | 18.67% | 21.03 | 33.37 | 14.13% |
| | AGCRN | 761K | 15.71 | 27.85 | 11.48% | 18.06 | 31.51 | 13.06% | 21.86 | 39.44 | 16.52% | 18.09 | 32.01 | 13.28% |
| | STGCN | 508K | 17.45 | 29.99 | 12.42% | 19.55 | 33.69 | 13.68% | 23.21 | 41.23 | 16.32% | 19.67 | 34.14 | 13.86% |
| | GWNET | 311K | 15.24 | 25.13 | 9.86% | 17.74 | 29.51 | 11.70% | **21.56** | **36.82** | 15.13% | **17.74** | 29.62 | 11.88% |
| | ASTGCN | 2.2M | 19.56 | 31.33 | 12.18% | 24.13 | 37.95 | 15.38% | 30.96 | 49.17 | 21.98% | 23.70 | 37.63 | 15.65% |
| | STTN | 114K | 16.22 | 26.22 | 10.63% | 18.76 | 30.98 | 12.80% | 22.62 | 39.09 | 16.14% | 18.69 | 31.11 | 12.82% |
| | STGODE | 729K | 16.75 | 28.04 | 11.00% | 19.71 | 33.56 | 13.16% | 23.67 | 42.12 | 16.58% | 19.55 | 33.57 | 13.22% |
| | DSTAGNN | 3.9M | 18.13 | 28.96 | 11.38% | 21.71 | 34.44 | 13.93% | 27.51 | 43.95 | 19.34% | 21.82 | 34.68 | 14.40% |
| | DGCRN | 243K | 15.34 | 25.35 | 10.01% | 18.05 | 30.06 | 11.90% | 22.06 | 37.51 | 15.27% | 18.02 | 30.09 | 12.07% |
| | D²STGNN | 406K | **14.92** | **24.95** | **9.56%** | **17.52** | **29.24** | **11.36%** | 22.62 | 37.14 | **14.86%** | 17.85 | **29.51** | **11.54%** |
| GBA | HL | – | 32.57 | 48.42 | 22.78% | 53.79 | 77.08 | 43.01% | 92.64 | 126.22 | 92.85% | 56.44 | 79.82 | 48.87% |
| | LSTM | 98K | 20.38 | 33.34 | 15.47% | 27.56 | 43.57 | 23.52% | 39.03 | 60.59 | 37.48% | 27.96 | 44.21 | 24.48% |
| | DCRNN | 373K | 18.71 | 30.36 | 14.72% | 23.06 | 36.16 | 20.45% | 29.85 | 46.06 | 29.93% | 23.13 | 36.35 | 20.84% |
| | AGCRN | 777K | 18.31 | 30.24 | 14.27% | 21.27 | 34.72 | 16.89% | **24.85** | **40.18** | 20.80% | 21.01 | 34.25 | 16.90% |
| | STGCN | 1.3M | 21.05 | 34.51 | 16.42% | 23.63 | 38.92 | 18.35% | 26.87 | 44.45 | 21.92% | 23.42 | 38.57 | 18.46% |
| | GWNET | 344K | 17.85 | 29.12 | 13.92% | 21.11 | **33.69** | 17.79% | 25.58 | 40.19 | 23.48% | 20.91 | **33.41** | 17.66% |
| | ASTGCN | 22.3M | 21.46 | 33.86 | 17.24% | 26.96 | 41.38 | 24.22% | 34.29 | 52.44 | 32.53% | 26.47 | 40.99 | 23.65% |
| | STTN | 218K | 18.25 | 29.64 | 14.05% | 21.06 | 33.87 | 17.03% | 25.29 | 40.58 | 21.20% | 20.97 | 33.78 | 16.84% |
| | STGODE | 788K | 18.84 | 30.51 | 15.43% | 22.04 | 35.61 | 18.42% | 26.22 | 42.90 | 22.83% | 21.79 | 35.37 | 18.26% |
| | DSTAGNN | 26.9M | 19.73 | 31.39 | 15.42% | 24.21 | 37.70 | 20.99% | 30.12 | 46.40 | 28.16% | 23.82 | 37.29 | 20.16% |
| | DGCRN | 374K | 18.02 | 29.49 | 14.13% | 21.08 | 34.03 | 16.94% | 25.25 | 40.63 | 21.15% | 20.91 | 33.83 | 16.88% |
| | D²STGNN | 446K | **17.54** | **28.94** | **12.12%** | **20.92** | 33.92 | **14.89%** | 25.48 | 40.99 | **19.83%** | **20.71** | 33.65 | **15.04%** |
| GLA | HL | – | 33.66 | 50.91 | 19.16% | 56.88 | 83.54 | 34.85% | 98.45 | 137.52 | 71.14% | 59.58 | 86.19 | 38.76% |
| | LSTM | 98K | 20.02 | 32.41 | 11.36% | 27.73 | 44.05 | 16.49% | 39.55 | 61.65 | 25.68% | 28.05 | 44.38 | 17.23% |
| | DCRNN | 373K | 18.41 | 29.23 | 10.94% | 23.16 | 36.15 | 14.14% | 30.26 | 46.85 | 19.68% | 23.17 | 36.19 | 14.40% |
| | AGCRN | 792K | **17.27** | 29.70 | 10.78% | **20.38** | 34.82 | **12.70%** | **24.59** | 42.59 | **16.03%** | **20.25** | 34.84 | **12.87%** |
| | STGCN | 2.1M | 19.86 | 34.10 | 12.40% | 22.75 | 38.91 | 14.11% | 26.70 | 45.78 | 17.00% | 22.64 | 38.81 | 14.17% |
| | GWNET | 374K | 17.28 | **27.68** | **10.18%** | 21.31 | **33.70** | 13.02% | 26.99 | **42.51** | 17.64% | 21.20 | **33.58** | 13.18% |
| | ASTGCN | 59.1M | 21.89 | 34.17 | 13.29% | 29.54 | 45.01 | 19.36% | 39.02 | 58.81 | 29.23% | 28.99 | 44.33 | 19.62% |
| | STGODE | 841K | 18.10 | 30.02 | 11.18% | 21.71 | 36.46 | 13.64% | 26.45 | 45.09 | 17.60% | 21.49 | 36.14 | 13.72% |
| | DSTAGNN | 66.3M | 19.49 | 31.08 | 11.50% | 24.27 | 38.43 | 15.24% | 30.92 | 48.52 | 20.45% | 24.13 | 38.15 | 15.07% |
| CA | HL | – | 30.72 | 46.96 | 20.43% | 51.56 | 76.48 | 37.22% | 89.31 | 125.71 | 76.80% | 54.10 | 78.97 | 41.61% |
| | LSTM | 98K | 19.04 | 31.28 | 13.19% | 26.49 | 42.63 | 19.57% | 38.22 | 60.29 | 30.28% | 26.89 | 43.11 | 20.16% |
| | DCRNN | 373K | 17.55 | 28.21 | 12.68% | 21.79 | 34.27 | 16.67% | 28.56 | 44.34 | 23.84% | 21.87 | 34.41 | 17.06% |
| | STGCN | 4.5M | 18.99 | 32.37 | 14.84% | 21.37 | 36.46 | **16.27%** | **24.94** | **42.59** | **19.74%** | 21.33 | 36.39 | **16.53%** |
| | GWNET | 469K | **17.14** | **27.81** | **12.62%** | 21.68 | **34.16** | 17.14% | 28.58 | 44.13 | 24.24% | 21.72 | **34.20** | 17.40% |
| | STGODE | 1.0M | 17.57 | 29.91 | 13.91% | **20.98** | 36.62 | 16.88% | 25.46 | 45.99 | 21.00% | **20.77** | 36.60 | 16.80% |

perform worst, since they only consider temporal dependencies and ignore the spatial correlations present in traffic data. The RNN-based method AGCRN and the TCN-based method GWNET generally outperform their predecessors, DCRNN and STGCN, on the SD, GBA, and GLA datasets. Even when compared to recent works like DGCRN and D²STGNN, their performance remains highly promising. This achievement can be attributed to their utilization of an adaptive adjacency matrix, which serves as a learnable fully-connected graph and significantly enhances model capacity. We also note that ASTGCN and DSTAGNN do not perform well on all datasets, which may be attributed to the model's large number of parameters, leading to potential overfitting issues. Notably, DGCRN and D²STGNN demonstrate impressive performance on the datasets of SD and GBA, validating the effectiveness of considering the dynamic characteristics of spatial topology. However, their complex model designs prevent them from scaling to larger datasets: GLA and CA. For the performance on the CA dataset, our results show that STGCN and STGODE can outperform GWNET on some of

Table 3: Efficiency comparisons. BS: batch size set during training. Train: training time (in seconds) per epoch. Infer: inference time (in seconds) on the validation set. Total: total training time (in hours). Note that the total training time is also influenced by the spent number of epochs.

| Method | SD | | | | GBA | | | | GLA | | | | CA | | | |
|---|---|---|---|---|---|---|---|---|---|---|---|---|---|---|---|---|
| | BS | Train | Infer | Total | BS | Train | Infer | Total | BS | Train | Infer | Total | BS | Train | Infer | Total |
| LSTM | 64 | 21 | 6 | 1 | 64 | 115 | 17 | 4 | 64 | 188 | 29 | 6 | 32 | 415 | 61 | 13 |
| DCRNN | 64 | 867 | 150 | 28 | 64 | 1,816 | 319 | 59 | 43 | 2,491 | 435 | 81 | 19 | 4,845 | 851 | 158 |
| AGCRN | 64 | 92 | 15 | 3 | 64 | 536 | 83 | 17 | 45 | 1,413 | 245 | 46 | – | – | – | – |
| STGCN | 64 | 53 | 16 | 2 | 64 | 160 | 54 | 6 | 64 | 268 | 86 | 10 | 64 | 701 | 206 | 25 |
| GWNET | 64 | 97 | 14 | 3 | 64 | 483 | 66 | 15 | 64 | 1,028 | 139 | 32 | 44 | 4,105 | 548 | 113 |
| ASTGCN | 64 | 128 | 19 | 4 | 45 | 1,126 | 147 | 35 | 17 | 3,060 | 393 | 77 | – | – | – | – |
| STTN | 64 | 208 | 26 | 6 | 7 | 1,758 | 197 | 50 | – | – | – | – | – | – | – | – |
| STGODE | 64 | 188 | 26 | 6 | 49 | 710 | 103 | 23 | 30 | 1,305 | 192 | 42 | 13 | 4,212 | 659 | 135 |
| DSTAGNN | 64 | 240 | 23 | 7 | 27 | 1,959 | 171 | 53 | 10 | 5,241 | 467 | 120 | – | – | – | – |
| DGCRN | 64 | 430 | 76 | 14 | 12 | 4,461 | 605 | 138 | – | – | – | – | – | – | – | – |
| D$^2$STGNN | 45 | 563 | 69 | 14 | 4 | 5,885 | 796 | 148 | – | – | – | – | – | – | – | – |

the metrics. However, they require 2 to $10\times$ more parameters compared to GWNET. In summary, we gain two key insights from the table: (1) Competitive methods introduced 3-4 years ago, namely GWNET and AGCRN, continue to perform well across evaluated datasets. Considering that they have been overlooked in some prior works, we believe that their significance should be acknowledged in future studies. (2) We observe that baselines span a wide range of parameter scales. To ensure fairness, evaluating models with comparable parameters is advisable in future research, similar to practices in computer vision and natural language processing.

## 5.3 Efficiency Comparisons

We summarize the efficiency comparisons in Table 3 with the following observations. LSTM is the fastest method due to its simple model architecture. Apart from this, TCN-based approaches like STGCN and GWNET are generally faster than all other baselines, thanks to the parallel computation of temporal convolution operations. They also demonstrate good scalability on large-scale datasets, as evidenced by the large batch size settings for the CA dataset. Among the RNN-based approaches, DCRNN and AGCRN are slower than TCN-based methods, and AGCRN exhibits significantly better speed than DCRNN due to its utilization of an MLP-like decoder instead of a recurrent decoder. On the other hand, DGCRN and D$^2$STGNN, despite their strong performance, suffer from long training and inference times. This is due to their complex model architectures that generate a substantial number of intermediate hidden states and thus require significant memory storage. Consequently, a small batch size must be utilized, leading to prolonged training duration. Moreover, the CA dataset poses significant challenges for existing traffic forecasting models, as only half of the selected baselines are capable of running on it. This outcome underscores the importance of developing scalable traffic forecasting models in future research.

## 6 Future Opportunities & Limitations

To facilitate accurate, efficient, and scalable traffic forecasting research, we introduce LargeST as a new benchmark dataset. It encompasses a total of 8,600 sensors, with each sensor containing 5 years of data and comprehensive metadata. According to the thorough data analysis and extensive experiment results, we highlight the following opportunities in future research.

- **The utilization of spatial, temporal, and metadata features.** Based on our data analysis, we have discovered positive correlations between traffic flow readings and regional distribution, temporal factors such as day of week and month in year, as well as metadata information like highway categories and lanes number. Future studies may consider incorporating this knowledge to enhance the accuracy and interpretability of traffic forecasting models.

- **A valuable testbed for the challenges of temporal distribution shifts.** Encompassing a comprehensive five-year temporal period, including the years 2020 and 2021 characterized by the global COVID-19 pandemic, our dataset offers a unique lens into temporal distribution shifts or out-of-distribution challenges. For example, researchers exploring the effects of extraordinary events on forecasting models can leverage our dataset as a testing ground to develop strategies for handling abrupt distribution shifts.

- **The development of simple yet effective methods.** After analyzing Tables 2 and 3, it becomes apparent that while the proposed methods demonstrate increasing accuracy in recent years, their models also become increasingly complex, which has a significant impact on their efficiency and scalability when applied to larger sensor networks. Therefore, there is a critical need to develop simple yet effective methods in traffic forecasting, to enable their practical implementation and deployment in real-world applications.

- **The development of foundation forecasting models.** Recently, developing foundation models has raised a surge of interest in a variety of domains, such as ChatGPT in natural language processing and Segment Anything in computer vision. With billions of curated data points, our dataset may serve as an invaluable resource for training foundation models in the fields of traffic forecasting or time series forecasting.

Our work offers a new traffic forecasting benchmark to facilitate research in the field, but it has limitations in terms of the dataset generalizability, as data analysis and all experiments were conducted in the areas of California. Another limitation of the LargeST dataset is associated with inaccuracies and missing data in the sensor readings. These issues can arise from factors such as signal interruptions and other unforeseen circumstances.

## Acknowledgments and Disclosure of Funding

This work by the authors is partly supported by the Advanced Research and Technology Innovation Centre (ARTIC), the National University of Singapore under Grant (project number: A-8000969-00-00). This work is also supported by Guangzhou Municipal Science and Technology Project 2023A03J0011, and the National Natural Science Foundation of China (No. 62372402).

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

# A    Dataset Documentation

We organize the dataset documentation based on the template of datasheets for datasets [9].

## A.1    Motivation

**For what purpose was the dataset created? Was there a specific task in mind? Was there a specific gap that needed to be filled? Please provide a description.**

This dataset is created for comprehensively evaluating the accuracy, efficiency, and scalability of current and future traffic forecasting models in large-scale scenarios. Current public benchmarks in traffic forecasting may not be applicable to practical scenarios due to the following limitations. First, the limited sizes of existing datasets may not reflect the real-world scale of traffic networks. Second, the temporal coverage of these datasets is typically short, posing hurdles in studying long-term patterns and acquiring sufficient samples for training deep models. Third, these datasets often lack adequate metadata for sensors, which compromises the reliability and interpretability of the data.

**Who created the dataset (e.g., which team, research group) and on behalf of which entity (e.g., company, institution, organization)?**

This dataset was created by Xu Liu, Yutong Xia, Yuxuan Liang, Junfeng Hu, Yiwei Wang, Lei Bai, Chao Huang, Zhenguang Liu, Bryan Hooi, and Roger Zimmermann. The authors are researchers affiliated with the National University of Singapore, Hong Kong University of Science and Technology (Guangzhou), Shanghai AI Laboratory, University of Hong Kong, and Zhejiang University.

**Who funded the creation of the dataset? If there is an associated grant, please provide the name of the grantor and the grant name and number.**

This work by the authors is partly supported by the Advanced Research and Technology Innovation Centre (ARTIC), the National University of Singapore under Grant (project number: A-8000969-00-00). This work is also supported by Guangzhou Municipal Science and Technology Project 2023A03J0011, and the National Natural Science Foundation of China (No. 62372402).

## A.2    Composition

**What do the instances that comprise the dataset represent (e.g., documents, photos, people, countries)? Are there multiple types of instances (e.g., movies, users, and ratings; people and interactions between them; nodes and edges)? Please provide a description.**

There are three types of data in our dataset. (1) The traffic flow readings obtained from loop detectors (sensors) installed in California state highway system. (2) The metadata of sensors. (3) An adjacency matrix that describes the graph topology.

**How many instances are there in total (of each type, if appropriate)?**

Our dataset has a total number of 8,600 sensors with a 5-year time coverage and includes 9 features for each sensor. It also possesses a sensor graph with 8,600 nodes and 201,363 edges.

**Does the dataset contain all possible instances or is it a sample (not necessarily random) of instances from a larger set? If the dataset is a sample, then what is the larger set? Is the sample representative of the larger set (e.g., geographic coverage)? If so, please describe how this representativeness was validated/verified.**

Our dataset is a sample of instances from a larger set in both spatial and temporal aspects. In terms of spatial coverage, the larger set includes all operational sensors in the California Department of Transportation Performance Measurement System (PeMS) [6]. As for the temporal aspect, the larger set consists of all traffic readings since the creation of the PeMS system. We consider our sample to be the representative of the larger set, as we specifically select sensors labeled as "mainline", which

---

[6]https://pems.dot.ca.gov

provides the overall traffic conditions throughout the entire system. Also, we include 5 years of recent data to ensure sufficient time coverage.

**What data does each instance consist of? "Raw" data (e.g., unprocessed text or images) or features? In either case, please provide a description.**

The traffic readings are just numerical values. The metadata of sensors consists of their coordinates, county, district in PeMS, the highway they are located on, directions of travel, and the number of lanes. The adjacency matrix records the connections between sensors.

**Is there a label or target associated with each instance? If so, please provide a description.**

No, traffic forecasting can be considered as a self-supervised learning task.

**Is any information missing from individual instances? If so, please provide a description, explaining why this information is missing (e.g., because it was unavailable).**

Yes, there are additional meta features associated with the sensors, such as the cities in which they are located. However, due to a high rate of missing data for these features, we have opted to ignore them in our analysis.

**Are relationships between individual instances made explicit (e.g., users' movie ratings, social network links)? If so, please describe how these relationships are made explicit.**

Yes, we use an adjacency matrix to denote the connections between sensors.

**Are there recommended data splits (e.g., training, development/validation, testing)? If so, please provide a description of these splits, explaining the rationale behind them.**

Yes, we chronologically split the data into train, validation, and test sets, with a ratio of 6:2:2.

**Are there any errors, sources of noise, or redundancies in the dataset? If so, please provide a description.**

Yes, the sensor readings are never perfectly accurate or sometimes missing due to unexpected factors, such as signal interruption.

**Is the dataset self-contained, or does it link to or otherwise rely on external resources (e.g., websites, tweets, other datasets)?**

Yes, it is self-contained.

**Does the dataset contain data that might be considered confidential (e.g., data that is protected by legal privilege or by doctor–patient confidentiality, data that includes the content of individuals' non-public communications)? If so, please provide a description.**

No, all our data are from a publicly available data source, i.e., PeMS.

**Does the dataset contain data that, if viewed directly, might be offensive, insulting, threatening, or might otherwise cause anxiety? If so, please describe why.**

No, all our data are numerical.

### A.3 Collection Process

**How was the data associated with each instance acquired? Was the data directly observable (e.g., raw text, movie ratings), reported by subjects (e.g., survey responses), or indirectly inferred/derived from other data (e.g., part-of-speech tags, model-based guesses for age or language)?**

We source the data from the California Department of Transportation (CalTrans) Performance Measurement System (PeMS). The data are directly observable.

**What mechanisms or procedures were used to collect the data (e.g., hardware apparatuses or sensors, manual human curation, software programs, software APIs)? How were these mechanisms or procedures validated?**

We initiate the dataset construction process by downloading a sensor list from the PeMS official website. This comprehensive list encompasses a majority of sensor meta-information, excluding the coordinates (longitude and latitude). Subsequently, we utilize the sensor IDs from this list to access both the coordinate data and historical observations of specific sensors via the website's APIs. Once we collect and store all the necessary sensor information, we engage in a vital data cleaning process, elaborated in Section A.4. The procedure for forming the adjacency matrix of the dataset is outlined in the main paper. This is basically how the CA dataset is built. We then construct the other three datasets, namely GLA, GBA and SD, through the selection of sensors positioned within these specific geographic areas.

**If the dataset is a sample from a larger set, what was the sampling strategy (e.g., deterministic, probabilistic with specific sampling probabilities)?**

The sampling strategy is deterministic.

**Who was involved in the data collection process (e.g., students, crowdworkers, contractors) and how were they compensated (e.g., how much were crowdworkers paid)?**

Our code collects publicly available data, which is free.

**Over what timeframe was the data collected? Does this timeframe match the creation timeframe of the data associated with the instances (e.g., recent crawl of old news articles)? If not, please describe the timeframe in which the data associated with the instances was created.**

The data are collected in 2023. This timeframe matches the creation timeframe of the data.

**Were any ethical review processes conducted (e.g., by an institutional review board)?**

No, such processes are unnecessary in our case.

### A.4   Preprocessing/cleaning/labeling

**Was any preprocessing/cleaning/labeling of the data done (e.g., discretization or bucketing, tokenization, part-of-speech tagging, SIFT feature extraction, removal of instances, processing of missing values)? If so, please provide a description.**

Yes, to ensure our dataset represents the overall traffic conditions throughout the entire system, we specifically select sensors labeled as "mainline". We also exclude sensors that lack coordinate information or are extremely far away from other sensors.

**Was the "raw" data saved in addition to the preprocessed/cleaned/labeled data (e.g., to support unanticipated future uses)? If so, please provide a link or other access point to the "raw" data.**

The raw data are available in PeMS. The link is: `https://pems.dot.ca.gov`.

**Is the software that was used to preprocess/clean/label the data available? If so, please provide a link or other access point.**

No.

### A.5   Uses

**Has the dataset been used for any tasks already? If so, please provide a description.**

The dataset is used in this paper for the traffic forecasting task.

**Is there a repository that links to any or all papers or systems that use the dataset? If so, please provide a link or other access point.**

No, but we may create one in the future.

**What (other) tasks could the dataset be used for?**

Traffic data imputation.

**Is there anything about the composition of the dataset or the way it was collected and preprocessed/cleaned/labeled that might impact future uses?**

We believe that our dataset will not encounter usage limit.

**Are there tasks for which the dataset should not be used? If so, please provide a description.**

No, users could use our dataset in any task as long as it does not violate laws.

## A.6 Distribution

**Will the dataset be distributed to third parties outside of the entity (e.g., company, institution, organization) on behalf of which the dataset was created? If so, please provide a description.**

No, it will always be held on GitHub.

**How will the dataset will be distributed (e.g., tarball on website, API, GitHub)? Does the dataset have a digital object identifier (DOI)?**

The instructions for building LargeST are available at: `https://github.com/liuxu77/LargeST`. The dataset does not have a digital object identifier currently.

**When will the dataset be distributed?**

On June 14, 2023.

**Will the dataset be distributed under a copyright or other intellectual property (IP) license, and/or under applicable terms of use (ToU)? If so, please describe this license and/or ToU, and provide a link or other access point to.**

Our benchmark dataset is released under a CC BY-NC 4.0 International License: `https://creativecommons.org/licenses/by-nc/4.0`. Our code implementation is released under the MIT License: `https://opensource.org/licenses/MIT`.

**Have any third parties imposed IP-based or other restrictions on the data associated with the instances? If so, please describe these restrictions, and provide a link or other access point to, or otherwise reproduce, any relevant licensing terms, as well as any fees associated with these restrictions.**

Yes, for commercial use, please check the website: `https://pems.dot.ca.gov`.

**Do any export controls or other regulatory restrictions apply to the dataset or to individual instances? If so, please describe these restrictions, and provide a link or other access point to, or otherwise reproduce, any supporting documentation.**

No.

## A.7 Maintenance

**Who will be supporting/hosting/maintaining the dataset?**

The authors of the paper.

**How can the owner/curator/manager of the dataset be contacted (e.g., email address)?**

Please contact this email address: liuxu12@u.nus.edu

**Is there an erratum? If so, please provide a link or other access point.**

Users can use GitHub to report issues or bugs.

**Will the dataset be updated (e.g., to correct labeling errors, add new instances, delete instances)? If so, please describe how often, by whom, and how updates will be communicated to dataset consumers (e.g., mailing list, GitHub)?**

Yes, the authors will actively update the code and data on GitHub.

**If the dataset relates to people, are there applicable limits on the retention of the data associated with the instances (e.g., were the individuals in question told that their data would be retained for a fixed period of time and then deleted)? If so, please describe these limits and explain how they will be enforced.**

The dataset does not relate to people.

**Will older versions of the dataset continue to be supported/hosted/maintained? If so, please describe how. If not, please describe how its obsolescence will be communicated to dataset consumers.**

Yes, we will provide the information on GitHub.

**If others want to extend/augment/build on/contribute to the dataset, is there a mechanism for them to do so? If so, please provide a description. Will these contributions be validated/verified? If so, please describe how. If not, why not? Is there a process for communicating/distributing these contributions to dataset consumers? If so, please provide a description.**

Yes, we welcome users to submit pull requests on GitHub, and we will actively validate the requests.

## B   More Dataset Information

This part endeavors to provide comprehensive details of the proposed LargeST benchmark dataset. Table 4 lists the main statistics of the sub-datasets in LargeST. The traffic flow data are provided in .h5 format, and the adjacency matrices are provided in .npy format. Table 5 shows the node metadata available in our dataset. The metadata file is provided in .csv format. Table 6 presents an overview of the statistical properties of two key features, namely highway categories and the number of lanes, which have been analyzed in the main paper.

Table 4: Dataset statistics. Degree: average node degree. Density: graph density.

| Dataset | Nodes | Edges | Degree | Density | Time Range | Sampling Rate | Time Frames |
|---------|-------|-------|--------|---------|------------|---------------|-------------|
| CA | 8,600 | 201,363 | 23.4 | 0.0027 | 01/01/2017 − 12/31/2021 | 5 minutes | 525,888 |
| GLA | 3,834 | 98,703 | 25.7 | 0.0067 | 01/01/2017 − 12/31/2021 | 5 minutes | 525,888 |
| GBA | 2,352 | 61,246 | 26.0 | 0.0111 | 01/01/2017 − 12/31/2021 | 5 minutes | 525,888 |
| SD | 716 | 17,319 | 24.2 | 0.0338 | 01/01/2017 − 12/31/2021 | 5 minutes | 525,888 |

## C   More Experimental Settings

We incorporate a total of 12 representative baselines in this study and comprehensively evaluate them on the LargeST benchmark dataset. In our GitHub repository, for each of the baseline, we provide the concrete model and training configurations in a .sh file. Users can directly execute this file to reproduce the reported experimental results. To ensure a fair efficiency comparison, we run all baseline models on the same GPU and measure their training time per epoch, inference time on the validation set, and total training hours.

Moreover, our codebase follows a modular design, allowing users to seamlessly add their custom models by following the established structure and guidelines. This modular approach enhances the codebase's extensibility and adaptability, and encourages experimentation with different model architectures and techniques. In the future, we will keep monitoring new research advancements in the field and actively summarize them in the repository.

Table 5: Metadata description.

| Attribute | Description | Possible Range of Values |
|---|---|---|
| ID | The identifier of a sensor in PeMS | A number with 6 to 9 digits |
| Lat | The latitude of a sensor | Real number |
| Lng | The longitude of a sensor | Real number |
| District | The district of a sensor in PeMS | 3, 4, 5, 6, 7, 8, 10, 11, 12 |
| County | The county of a sensor in California | Sacramento, Yolo, El Dorado, Placer, Nevada, Sutter, Contra Costa, Napa, Solano, Santa Cruz, Santa Clara, Alameda, San Benito, Marin, Sonoma, San Francisco, San Mateo, Monterey, Santa Barbara, San Luis Obispo, Kern, Kings, Fresno, Madera, Tulare, Los Angeles, Ventura, San Bernardino, Riverside, San Joaquin, Merced, Stanislaus, San Diego, Orange |
| Fwy | The highway where a sensor is located | I5, US50, SR51, SR65, I80, SR99, SR4, SR12, SR17, SR24, SR25, SR37, SR85, SR87, SR92, US101, SR237, SR238, SR242, I280, I580, I680, I780, I880, I980, SR68, SR41, SR58, SR168, SR180, SR198, SR2, I10, SR14, SR23, SR47, SR57, SR60, SR71, SR91, I105, I110, SR118, SR134, SR170, I210, I405, I605, I710, I15, I215, SR120, SR132, SR219, I8, SR11, SR52, SR54, SR56, SR78, SR94, SR125, SR163, I805, I905, SR22, SR55, SR73, SR133, SR142, SR241, SR261 |
| Lane | The number of lanes where a sensor is located | 1, 2, 3, 4, 5, 6, 7, 8 |
| Type | The type of a sensor | Mainline |
| Direction | The direction of the highway | N, S, E, W |

Table 6: Statistic of two meta features.

| Dataset | Highway Categories | | | Number of Lanes | | |
|---|---|---|---|---|---|---|
| | Interstate | U.S. | State | 1 to 2 | 3 to 4 | >=5 |
| CA | 4,322 | 1,234 | 3,044 | 1,347 | 5,679 | 1,574 |
| GLA | 2,316 | 184 | 1,334 | 280 | 2,769 | 785 |
| GBA | 1,108 | 731 | 513 | 349 | 1,459 | 544 |
| SD | 516 | – | 200 | 73 | 449 | 194 |

