# OpenReview forum: "LargeST: A Benchmark Dataset for Large-Scale Traffic Forecasting"
_NeurIPS.cc/2023/Track/Datasets_and_Benchmarks — NeurIPS 2023 Datasets and Benchmarks Poster_

### Official Review · Reviewer_G2Gc · 2023-07-21
**This article introduces a traffic forecasting benchmark dataset LargeST, which includes the comprehensive metadata from 8,600 sensors with the 5-year coverage period.**

**Rating:** 8
**Confidence:** 4
**Correctness:** The dataset is constructed in a sound…
**Clarity:** The paper is well-written.

**Strengths:**

+ A large benchmark dataset for traffic forecasting model is published, to be available for researchers to develop new traffic forecasting models.
+ The comprehensive analysis for the dataset can help readers better understand the dataset.
+ Extensive experiments are conducted to show the usability of this dataset across different models.
+ The well-thought discussions for presenting the future opportunities for utilizing such a dataset.

**Additional Feedback:**

Please see my comments above.

**Documentation:**

The sufficient details are provided.

**Limitations:**

Please see my comments above.

**Opportunities For Improvement:**

- Providing the storage format and the access method to this dataset.
- It is better to provide some solutions for processing the nodes missing the traffic flow. Will it degrade the quality of this dataset by including these empty nodes?

**Relation To Prior Work:**

This article clearly discussed its difference from previous contributions.


**Summary And Contributions:**

This article introduces a traffic forecasting benchmark dataset, LargeST, crafted from the California Department of Transportation Performance Measurement system. Such a dataset is much larger than all previous benchmarks, encompassing the data from a total of 8600 sensors, has a five-year temporal coverage, and includes rich node metadata. The data collection and organization are depicted and statistic analysis is provided. The experimental results are presented for validating the performance of 11 baselines on such a dataset.

---

> ### Author Response · Authors · 2023-08-20
>
> Dear reviewer, thank you very much for appreciating the data analysis and experimental parts in this study. Please find our responses below.
>
> **[C1] Providing the storage format and the access method to this dataset.**
>
> Thanks for pointing this out. The storage format of the datasets is stated in Section B of supplementary material. Concretely, the traffic flow data are provided in .h5 format, the adjacency matrices are provided in .npy format, and the metadata file is in .csv format. The users can simply employ the Pandas or NumPy libraries to read the data. We also have included the example code to read data on our GitHub website.
>
> **[C2] It is better to provide some solutions for processing the nodes missing the traffic flow. Will it degrade the quality of this dataset by including these empty nodes?**
>
> Thanks for asking this. In this study, we intend to maintain the data in its original state as much as possible, so that users have the discretion to decide whether to fill missing values or not. Moreover, there exist a number of tools/methods to fill the missing values in the time series data. For example, the interpolate() function in the Pandas library (https://pandas.pydata.org/docs/reference/api/pandas.DataFrame.interpolate.html) is a handy and simple tool to do this. We have made this part more clear in our revised paper (Line 150-155).

---

### Official Review · Reviewer_4UAT · 2023-07-21

**Rating:** 6
**Confidence:** 4
**Correctness:** Yes
**Clarity:** Yes

**Strengths:**

1. The paper is readable and well-organized, I enjoy reading the paper.
2. The proposed dataset is solid, containing records from 2017 to 2021.
3. The authors conduct experiments with multiple SOTA models to demonstrate the effectiveness of their proposed dataset.

**Additional Feedback:**

NULL

**Documentation:**

Yes

**Limitations:**

1. The author should state the definition of "traffic forecasting" clearly. Several networking solutions also mention traffic classification/prediction. Differently, in this paper, the authors typically discuss car traffic (if I understand correctly) instead of network traffic. I suggest adding a clear statement of "traffic" in the introduction to help readers identify this work from networking traffic.
2. I also hope the authors make a further discussion about the following questions:
a. Is the older the records maintained by the dataset the better? People's behaviors may change over time, and the old records may mislead the model. For example, during the covid-19, people may work from home, which results in lower traffic compared to 2017.
b. why the filter threshold r is 0.01? Is there any evidence to support this setting?

**Opportunities For Improvement:**

1. clear definition is needed.
2. Issues about the proper collection period and hyperparameter selection should be discussed.
(See my detailed explanation below)

**Relation To Prior Work:**

Yes

**Summary And Contributions:**

This paper creates a new dataset LargeST for car traffic prediction. LargeST is larger than previous datasets and maintains temporal information and node metadata. With LargeST, the authors analyze the performance of current prediction DL models, especially the
model based on GNN. Finally, the paper discusses the potential opportunities and directions for future research based on
LargeST.

---

> ### Author Response · Authors · 2023-08-20
>
> Dear reviewer, we would like to sincerely thank you for the time and effort put into reviewing our submission. We are grateful for your acknowledgement of the quality of our presentation and the valid experimental evaluations. Please find our responses below.
>
> **[C1] The author should state the definition of "traffic forecasting" clearly. I suggest adding a clear statement of "traffic" in the introduction to help readers identify this work from car traffic.**
>
> Thank you for this helpful comment. We have modified the expression in the introduction part to "road traffic forecasting" to reflect this point. Please feel free to check our revised paper (Line 1 and Line 18).
>
> **[C2] Is the older the records maintained by the dataset the better? People's behaviors may change over time, and the old records may mislead the model. For example, during the covid-19, people may work from home, which results in lower traffic compared to 2017.**
>
> Thanks for your insightful comment. This is actually a question about temporal distribution shifts or out-of-distribution issues. We argue that our dataset actually offers the unique opportunities to investigate distribution shifts over extended periods, a capability not attainable with previously released datasets. For example, if we would like to predict the traffic conditions in Oct. 2019, is it truly necessary to incorporate training samples from Oct. 2017? Furthermore, when encountering abrupt distribution shifts, such as those induced by events like COVID-19, it becomes essential to examine how the current forecasting models will respond and how to develop effective strategies for mitigating such challenges and changes. We have added this part into Section 6 of the revised paper (Line 307-312), please feel free to check it.
>
> **[C3] Why the filter threshold r is 0.01? Is there any evidence to support this setting?**
>
> Thanks for asking this. To the best of our knowledge, starting from DCRNN [1], it is a well-established practice in the research field of traffic forecasting to use a small threshold to prune weak node connections. In the experimental study of our paper, we have opted to use a threshold of 0.01 to eliminate weak node connections, while ensuring an adequate number of edges are retained. However, the selection of this threshold value is adaptable, and users may employ alternative values that align with their specific needs. We have clarified this point in the revised paper (Line 145-148).
>
> [1] Diffusion convolutional recurrent neural network: Data-driven traffic forecasting. ICLR 2018.
>
> Thank you again for your detailed review and insightful feedback, which helps us improve the quality of the paper. Please let us know if you have any other comments/questions.

---

> > ### Comment · Reviewer_4UAT · 2023-08-21
> >
> > The authors have addressed my questions.
> >
> > I have improved this paper's rating.

---

> > > ### Author Response · Authors · 2023-08-21
> > >
> > > Dear reviewer, thank you for taking the time to review our responses and for your insightful feedback throughout the review process.

---

### Official Review · Reviewer_4GMz · 2023-07-21
**Review for LargeST**

**Rating:** 6
**Confidence:** 4
**Clarity:** Yes, this paper is well written.

**Strengths:**

1. This paper highlights the limitations in terms of datasets in the field of traffic prediction, emphasizing the urgent need to address them.
2. This paper is well-written and easy to read. Each figure is beautifully drawn as well.
3. This paper provides a relatively large-scale traffic prediction dataset and includes benchmarks and reproducible code.

**Additional Feedback:**

N/A.

**Correctness:**

As I understand it, the original dataset does not belong to the authors of the paper. Therefore, I ask two main questions:
- Are secondary releases of datasets permissible? Is the description of Sec4.3 sufficient?
- Can you provide a more detailed reason why only some of the nodes were selected?

**Documentation:**

Good.

**Ethics:**

N/A.

**Limitations:**

1. Firstly, I believe that the scale of LargeST can only be considered relatively large compared to other datasets, but it cannot be regarded as truly large. While I know that this paper provides a relatively large dataset of 8000 nodes, it falls significantly short when compared to truly large-scale graph datasets, such as OGB (over 132K nodes).
2. The coverage of this paper is limited. Similar tasks to traffic prediction, such as weather forecasting, parking availability prediction, electricity prediction, and others, can be collectively referred to as spatio-temporal prediction. However, in these fields, the datasets provided are much larger than the LargeST. For example, the coverage of this paper[1] extends far beyond 8000 nodes. The experiments in this paper[2] were conducted on a dataset of approximately 2000 nodes.
3. Indeed, the metadata provided in these tasks is not rich. Some important metadata, such as weather conditions, holidays, and other factors, have not been considered in the dataset.
4. This paper does not demonstrate sufficient baselines.


[1] Skilful nowcasting of extreme precipitation with NowcastNet. Nature 2023.
[2] Semi-Supervised Hierarchical Recurrent Graph Neural Network for City-Wide Parking Availability Prediction. AAAI 2020.

**Opportunities For Improvement:**

1. The baseline implementation of this paper is not sufficiently refined. On one hand, I hope the authors can enrich the baseline or provide substantial evidence to explain why this result cannot be achieved. Besides, while spatiotemporal graph models are the primary approach for traffic flow prediction, Transformer-based methods are equally important.

2. The paper presents some visually appealing figures; however, I did not find any interesting conclusions/phenomenon, especially regarding the changes that occur as the dataset size increases.

**Relation To Prior Work:**

This paper provides comparisons of some commonly used traffic prediction datasets. However, to my knowledge, DiDi has previously provided larger-scale datasets, but due to policy reasons, they are not currently being released.

**Summary And Contributions:**

This paper presents a large-scale traffic prediction dataset called LargeST, which addresses three limitations in the field of traffic forecasting: size, temporal coverage, and node metadata. Additionally, the paper provides benchmark results of a spatiotemporal graph-based traffic prediction models on LargeST.

---

> ### Author Response · Authors · 2023-08-20
> **Official Comment by Authors (Part 1 / 2)**
>
> Dear reviewer, thank you for your thorough review and constructive feedback on our paper. We appreciate the time you have taken to provide valuable insights. Please find our responses below.
>
> **[C1] I hope the authors can enrich the baseline, e.g., Transformer-based methods.**
>
> Thanks for the comment. There is one attention-based baseline, i.e., ASTGCN, already in the paper. Following your suggestion, we have added one more Transformer-based method STTN [1] during the rebuttal stage. This paper is influential and has around 200 citations. Please note that this method cannot scale to the GLA and CA datasets, and is only able to run on SD and GBA. The results have been added to the Table 2 and 3 of the revised paper, and please feel free to check them.
>
> [1] Spatial-Temporal Transformer Networks for Traffic Flow Forecasting. Arxiv 2020.
>
> **[C2] The paper presents some visually appealing figures; however, I did not find any interesting conclusions/phenomena, especially regarding the changes that occur as the dataset size increases.**
>
> Thanks for raising this concern. In comparison to prior datasets in traffic forecasting, LargeST increases the data size in three dimensions: graph size, temporal coverage, and node metadata. Here we would like to list two changes that occur as the dataset size increases: (1) In Figures 2 (c) and (d), we delve into monthly patterns on a yearly scale, a level of analysis unattainable with prior datasets, which typically encompass less than six months of data. (2)  Figure 3 illustrates the correlations between traffic flow and two pivotal features. These two features, unavailable in prior datasets, contribute to a more comprehensive understanding of the data.
>
> **[C3] The scale of LargeST can only be considered relatively large compared to other datasets. LargeST falls significantly short when compared to truly large-scale graph datasets, such as OGB (over 132K nodes).**
>
> Thanks for the comment. Our statement in the paper is based on closely similar dataset types. However, the OGB dataset lacks a temporal dimension, which holds significant importance in traffic forecasting research, and thus we believe it is not directly comparable. On the other hand, it is evident from Table 1, LargeST consists of around four billion data points. From this perspective, LargeST's scale is not considerably smaller than that of OGB.
>
> **[C4] The coverage of this paper is limited. Similar tasks to traffic prediction, such as weather forecasting and parking availability prediction can be collectively referred to as spatio-temporal prediction. However, in these fields, the datasets provided are much larger than LargeST. Paper [1] extends far beyond 8000 nodes. Paper [2] is conducted on a dataset of 2000 nodes. [1] NowcastNet. Nature 2023. [2] City-Wide Parking Availability. AAAI 2020.**
>
> Thanks for raising this concern. We absolutely agree with your perspective that traffic forecasting, just like weather prediction and estimating parking availability, falls under the category of spatio-temporal predictions. However, we wish to clarify that we have not overly emphasized the significance of the size of our dataset when considering spatio-temporal predictions in general. In this study, our primary focus lies within the field of traffic forecasting research.
>
> Furthermore, we observe that the complete dataset referenced in [1] is not accessible to the public. This dataset carries certain restrictions and requires application, with permissions contingent upon the China Meteorological Administration. Additionally, the data in [2] has not been made open source. Thus we believe that our open dataset will be valuable for the community.
>
> **[C5] The metadata provided in these tasks is not rich. Some important metadata, such as weather conditions, holidays, and other factors, have not been considered in the dataset.**
>
> We greatly appreciate your insightful comment. While it is possible that integrating this additional information could improve performance, it is non-trivial to collect them in a short period. For example, we may need to find some reliable sources to acquire weather conditions in California spanning five years. Furthermore, ensuring alignment between the weather data and the specific sensor locations adds complexity to the task. Given the substantial workload entailed, we will leave this in our future work.

---

> > ### Author Response · Authors · 2023-08-20
> > **Official Comment by Authors (Part 2 / 2)**
> >
> > **[C6] Are secondary releases of datasets permissible? Is the description of Sec. 4.3 sufficient?**
> >
> > Thanks for asking this. Yes, we checked the PeMS official website and it is permissible to do secondary releases of datasets for research purposes. Moreover, most of the existing publicly datasets listed in Table 1 are sourced from PeMS. Thus, we believe the description in Section 4.3 is sufficient.
> >
> > **[C7] Can you provide a more detailed reason why only some of the nodes were selected?**
> >
> > Thank you for this question. The sensors in the PeMS system are labeled with different types. In this study, we specifically select sensors labeled as "mainline" for two primary reasons. First, these sensors can ensure that our dataset represents the overall traffic conditions throughout the entire system. Second, the other types of sensors generally register low traffic volumes. Predicting their future values might not yield as meaningful or beneficial insights compared to forecasting traffic volume on the mainline, where the flow often reaches hundreds of cars per 5 minutes. Furthermore, for data cleaning considerations, we exclude sensors that lack coordinate information or are extremely far away from other sensors.
> >
> > We are very grateful for your insightful comments, which have certainly given us an opportunity to fine-tune our messaging. Your expert knowledge is helping us to strengthen the manuscript significantly. Please let us know if you have any other comments/questions.

---

> ### Author Response · Authors · 2023-08-28
>
> Dear Reviewer 4GMz:
>
> Thank you so much for the precious review time and valuable comments. Following your suggestions, we have revised our paper and provided corresponding responses. We hope to further discuss with you whether or not your concerns have been addressed. Please let us know if you still have any unclear parts of our work.
>
> Best regards,
> Authors

---

> > ### Comment · Reviewer_4GMz · 2023-08-28
> >
> > Thanks for your reply. While there are some issues (such as scale) that are still worth discussing, I think this is a valuable resource paper. As a result, I ended up raising my score.

---

> > > ### Author Response · Authors · 2023-08-28
> > >
> > > Dear reviewer, thank you for taking the time to read our responses and for updating your review! We are glad to hear that you find our work valuable to the community.

---

### Official Review · Reviewer_M7Lz · 2023-07-25
**I think it's a good paper that fits Datasets and Benchmarks Track and has practical applications.**

**Rating:** 7
**Confidence:** 3
**Correctness:** Yes
**Clarity:** Yes, but it has some typos and spelli…

**Strengths:**

1. This paper is well-written and structured, with figures and tables that visually tell me the key information.

2. This paper fits the Datasets and Benchmarks Track, and as can be seen from the github link, the documentation is well written.

3. It solves a important problem in the field of traffic prediction and is fully original.

4. There is ample experimental evidence of its effectiveness.

**Additional Feedback:**

I have no more feedback.

**Documentation:**

Yes

**Limitations:**

No, the authors have not adequately addressed the limitations and potential negative societal impact of their work. Please add it.

**Opportunities For Improvement:**

1. Please add some images of the dataset as it was collected, and the results of the dataset visualisation.

2. There are some typos and spelling errors. Please check them carefully.

**Relation To Prior Work:**

Yes

**Summary And Contributions:**

The authors present a benchmark dataset called LargeST. The dataset contains 8600 sensors, covers a 5-year time span, and contains comprehensive sensor metadata. Using the LargeST dataset, the authors perform in-depth data analysis to extract data insights, benchmark well-known baseline models in terms of performance and efficiency, and identify challenges and opportunities for future research.

---

> ### Author Response · Authors · 2023-08-20
>
> Dear reviewer, thank you very much for appreciating the quality of our presentation and the comprehensiveness of our empirical evaluations. Please find our responses below.
>
> **[C1] Please provide more dataset visualization, fix typos and spelling errors in the paper, add limitations and social impact.**
>
> Thanks for your valuable advice. We have added dataset visualization in the supplementary file Section A.4 (Line 113-118). We have carefully checked for typos/spelling errors, and specifically pointed out our limitations in our revised paper (Line 324-328). Please feel free to check them.

---

### Official Review · Reviewer_nPQ9 · 2023-07-28
**Review of LargeST: A Benchmark Dataset for Large-Scale Traffic Forecasting**

**Rating:** 6
**Confidence:** 3
**Clarity:** The paper is nicely written and is ea…

**Strengths:**

- The paper makes a good effort to highlight the contribution of the dataset, in particular, the scale of the traffic information being significantly larger than existing works; this is clearly shown in Table 1.

- The exploratory analysis in Figure 2/3 provides insights into the traffic flow patterns, making a good contribution to an important problem.

**Additional Feedback:**

- I could not find a sufficient description of the dataset construction procedure based on the source data. The supplementary file merely contains a list of questions/checkpoints. I believe having a description would be helpful for other researchers to understand the data curation process better.

- While I appreciate the authors' effort in curating a large-scale traffic dataset, it remains unclear to what extent this dataset could benefit future research. Could the authors provide a more concrete discussion of some plausible research outcomes based on constructing this new dataset?

**Correctness:**

An opensource GitHub repository has been provided, and the dataset has been deposited on Kaggle for public access. The dataset construction procedure is described in detail in Section 4.1.

There are some details related to building the adjacency matrix of the sensor graph based on road network distances and OpenStreetMap data.

**Documentation:**

The GitHub repository provides a detailed description of accessing and using the dataset.

**Limitations:**

The authors discussed opportunities for future research near the end of the paper. The potential negative societal impact seems limited in light of the nature of this work.

**Opportunities For Improvement:**

- I felt there was a lack of interpretation of the experimental findings in Table 2. For instance, it would be helpful if a summary of the overall takeaways could be clearly stated.

- Although the experiments section focuses on comparing state-of-the-art spatiotemporal graph neural networks, it would have been nice if simpler methods such as graph embeddings (e.g., node2vec) could be incorporated/calibrated because these methods usually provide more direct graph-structural features.

- It was only stated until Page 4 that the authors are focusing on sensors measured in California and for four different metropolitan areas within California. This is not obvious based on the writing of the abstract and the introduction, which can be a bit misleading/overselling. I suggest the authors revise the abstract and introduction to make it clear that the dataset is constructed based in CA and for four metropolitan areas within CA. It is not immediately obvious whether the experiment findings would apply to a different US state or to a different country. This seems like an obvious limitation of the dataset, and it should be clearly discussed to avoid confusion.

**Relation To Prior Work:**

This work mainly differs from previous contributions in terms of the scale of the traffic flow dataset constructed.

**Summary And Contributions:**

This paper constructs a large-scale spatiotemporal traffic flow dataset, aimed toward making progress in the problem of traffic forecasting. The paper expands on several recent papers in this direction, such as Y. Li et al. (ICLR'17), and proposes a dataset whose scale is larger than previous works.

To summarize the contribution of this new dataset, it includes traffic networks for four metropolitan areas, and the size of the network ranges between a few hundred to several thousand nodes.
- The dataset includes traffic flow information, which is sourced from loop detectors measured by the California Department of Transportation's Performance Measurement System. The time range is from 2017 to 2021, including a total of over 500 thousand time frames of five-minute intervals.

- The dataset also provides metadata information, including the highway categories and the number of lanes across the four datasets.

Based on the construction of this new dataset, the paper first conducts exploratory data analysis to identify patterns in the data. Then, it provides a comprehensive evaluation of existing spatiotemporal graph neural networks for predicting traffic flow information. The experimental results highlight several recent architectures such as D$^2$STGNN and DGCRN as promising approaches for predicting future traffic information.

---

> ### Author Response · Authors · 2023-08-20
>
> Dear reviewer, we would like to sincerely thank you for the time and effort put into reviewing our submission. Your feedback is invaluable and helps enhance the quality of our paper. Please find our responses below.
>
> **[C1] Lack of interpretation of the experimental findings in Table 2. For instance, it would be helpful if a summary of the overall takeaways could be clearly stated.**
>
> Thanks for this valuable advice. In summary, we can gain two key insights from the table: (1) Competitive methods introduced 3-4 years ago, namely GWNET and AGCRN, continue to perform well across evaluated datasets. Considering that they have been overlooked in some prior work, we believe that their significance should be acknowledged in future studies. (2) We observe that baselines span a wide range of parameter scales. To ensure fairness, evaluating models with comparable parameters is advisable in future research, similar to practices in computer vision and natural language processing. We have added this part into our revised paper (Line 275-281) and please feel free to check it.
>
> **[C2] Clarification of LargeST is constructed based in California, and the experimental findings may not apply to different US states or different countries.**
>
> Thanks for raising this concern. As a quick note, from Figure 1 we can observe that LargeST is built based on sensors in California. And we agree with you: we have modified the abstract and introduction sections to directly reflect this point (Line 11 and Line 46). Also, we have added a limitation part into the revised paper (Line 324-328). Please feel free to check them.
>
> **[C3] The description of the dataset construction procedure is not sufficient.**
>
> Thanks for pointing this out. We have added more details on how the dataset was constructed in the supplementary material. Please check Appendix A.3 (Line 88-96) and A.4 (Line 113-118).
>
> **[C4] To what extent this dataset could benefit future research is unclear.**
>
> Thank you for this question. We believe that our dataset holds substantial potential to contribute to future research endeavors in traffic forecasting / time series forecasting for the following reasons: (1) Our dataset presents a robust platform for evaluating the scalability of future forecasting models. As we can observe in Table 3, there are only a few approaches that can scale to the CA dataset. (2) Recently, developing foundation models has raised a surge of interest in a variety of domains, such as ChatGPT in natural language processing and Segment Anything in computer vision. With billions of curated data points, our dataset may serve as an invaluable resource for training foundation models in the fields of traffic forecasting or time series forecasting. (3) Encompassing a comprehensive five-year temporal period, including the years 2020 and 2021 characterized by the global COVID-19 pandemic, our dataset offers a unique lens into temporal distribution shifts or out-of-distribution challenges. For example, researchers exploring the effects of extraordinary events on forecasting models can leverage our dataset as a testing ground to develop strategies for handling abrupt distribution shifts. We have added this part into the end of our revised paper (Line 307-312 and Line 319-323).
>
> Once again, thank you for your constructive feedback. We hope our revisions and clarifications address your concerns.

---

### Decision · Program_Chairs · 2023-09-22

**Decision:**

Accept (Poster)

**Comment:**

The authors introduce the LargeST dataset, a spatiotemporal traffic dataset covering four metropolitan areas across five years, which is larger and more detailed than prior works. The authors evaluate existing graph neural networks for predicting traffic flow.

The reviewers generally agree that this work makes a strong contribution to an important problem. Furthermore, the paper is well-written, and it overcomes the limitations of prior similar works.

The authors generally addressed the concerns of the reviewers during the rebuttal period.

I recommend acceptance, and I encourage the authors to ensure that all of the comments by the reviewers are addressed in the final version of the paper.